# Gaming Tree Based Evaluation Model for Badminton Tactic Benefit Analysis and Prediction

**Wenming Liu [1], Yifan Zhu [2], Wenxia Guo [3], Xinyuan Wang [4] and Songkun Yu [5,*]**

[1] Department of Sports Science, Zhejiang University, Hangzhou 310027, China; liuwenming@zju.edu.cn
[2] College of Computer Science, Zhejiang University, Hangzhou 310027, China; xtf_z@zju.edu.cn
[3] Department of Social Sports, Beijing University of Chemical Technology, Beijing 100013, China; guowenxia@buct.edu.cn
[4] Sports and Military Training Department, Zhejiang University of Technology, Hangzhou 310027, China; wwwwwxy0408@zjut.edu.cn
[5] Department of Public Physical and Art Education, Zhejiang University, Hangzhou 310027, China
[*] Correspondence: yusongkun@zju.edu.cn

**Abstract:** Badminton tactics refer to the techniques and strategies employed by players to win a match. Analyzing these tactics can help players improve their performance and outsmart their opponents. To study the tactics of top players, we use a gaming tree to analyze matches between two of the most powerful badminton players in history: Lin and Lee. By employing the Nash Equilibrium, we can discover the most beneficial strategies for both players, which reflect their most powerful techniques. Additionally, with the help of this gaming tree, we can precisely predict how players will implement their tactics. Empirical experimental results demonstrate that our proposed method not only evaluates and identifies each player's weaknesses and strengths but also has powerful capabilities to predict their tactics.

**Keywords:** tactic analysis; gaming tree; nash equilibrium; badminton analysis and prediction; computer-based analysis





## 1. Introduction

Badminton is one of the most popular sports in the world [1,2], and is also the racquet sport with the highest speed in the world [3], with its competitive form being two single/pair athletes completing tactical implementation quickly and accurately [4,5] on an 80 square meter court. As a result, tactical strategies (strategical strokes) are one of the key factors for winning badminton matches [6].

With the rapid development of information technology, advanced statistical methods such as data mining and artificial intelligence are commonly used in tactical analysis. Such methods can mine implicit information from performance data of competitions to provide decision support for coaches and athletes, which is an important aspect of tactical strategy research for sports such as basketball [7,8], football [9,10], tennis [11,12], table tennis [13,14], and badminton [15,16] and is also a promising direction for improvement and development of traditional performance diagnosis and evaluation methods [17,18]. In recent years, predicting match results (scores, wins/losses) using these methods has become a hot research topic [19,20]. Through prediction models, key tactical factors affecting match results can be analyzed. For example, Valero et al. [21] uses four data mining methods (including lazy learners, artificial neural networks, support vector machines, and decision trees) to evaluate classification-based and regression-based methods for predicting match results (home team win or lose) in Major League Baseball (MLB) regular season games over 10 years; Razali et al. [22] used Bayesian networks to predict home wins, away wins, and draws in the English Premier League; Karlis and Ntzoufras [23] constructed a bivariate

Poisson model to analyze scores between two teams, etc. However, there has been no prediction research on badminton so far.

Game theory studies how rational actors make decisions and the equilibrium of such decisions under the assumption of mutual interaction and influence among relevant actors. The prediction ability, actual behavior selection and optimal selection between the two sides of the game are the research focuses of game theory. Taking the concept of Nash Equilibrium as an instance. Nash Equilibrium describes a state where each player in a game has chosen a strategy, and no players can improve their outcome by changing their strategies while others keep their strategies unchanged. Thus, Nash Equilibrium helps find each player's optimal decision for scoring. Consequently, game theory has been widely applied in sports science, including sports teaching [24,25], tactical strategies [26–28], etc., while research on its prediction in the sports field has not yet been studied.

Badminton matches have strong confrontation and competition and have obvious interactive and interdependent characteristics of tactics and strategies between opponents, which is consistent with the research object characteristics of game theory. Therefore, this study intends to use the game tree method with game theory to explore the winning rate of badminton matches. By using the important badminton match data of recent years, this study constructs the gaming tree to model the tactic strategies of players in existing matches and proposes the evaluation model to predict the winning rate based on the given strokes.

## 2. Materials and Methods

### 2.1. Materials

In total, 29 matches of 2 of the most famous badminton players (i.e., Dan Lin and Chong Wei Lee) from 2006 to 2018 were selected. As men's single players, Lin has won nine major titles in the badminton world with 2 Olympic gold medals, while Lee was ranked first worldwide for 349 weeks including a 199-week streak. Thus, the selected matches between Lin and Lee present the highest-level badminton techniques and tactics and are worth analyzing. All match videos were from television relay or the Internet. Note that the two players have announced their retirement, and the selected matches are only used for analysis, i.e., no technical guidance is provided.

### 2.2. Observation Indices and Tactical Combination

Based on the studies by Butterworth and Turner [29] and Phomsoupha and Laffaye [2], we concluded the tactical observation indices including stroke technique, stroke placement, and rally results, to be as follows:

- Stroke technique: Serve, including short serve and long serve; Smash, an aggressive overhead shot with a downward trajectory; Clear, an overhead shot with a flat or rising trajectory towards the back of the opponent's court; Drop, is a smooth shot from above the head with a downward trajectory towards the front of the court; Net shot, denoting a precise shot from near the net, including the net drop, lob and kill; Drive, a powerful shot made at middle body height and in the middle of the court with a flat trajectory;
- Stroke placement: the start position and the target placement of each stroke. In this paper, the badminton court is evenly divided into 9 (3 × 3) grids, i.e., the combination of vertically three parts (front court, middle court, and back court) and horizontally three parts (left court, middle court, and right court);
- The rally results: scoring and losing.
- In fact, the speed of each stroke also contributes to the tactics. However, as the speed (including smash speed, clear shoot speed, etc.) of high-level players are almost the same (especially for Lin and Lee), the influence of the stroke speed for the rally results is not considered in this paper.

Based on the above, the tactical combination is composed of different stroke techniques and stroke placements of each stroke by two players, where each stroke has four attributes,

i.e., the start position, the applied technique, the target placement, and the final result for the rally that this stroke belongs to.

### 2.3. Tactical Frequency and Scoring Rate Algorithm

In this study, the attributes of different strokes for each rally are computed first. Let P be a binary set that represent the stroking player, $X$ be a set of stroke techniques, $Y$ be a set of stroke placements, S be a set of strokes and $D$ be a set of descriptive vectors of all rallies. A stroke $s_k^i \in S$ for the $k^{th}$ stroke in the $i^{th}$ rally can be denoted as $(p_k^i, x_k^i, y_k^i, y_{k+1}^i)$, where $p_k^i$ is the player for this stroke, $x_k^i$ is the applied technique, $y_k^i$ is the start place for this stroke and the destination of this stroke is denoted as $y_{k+1}^i$. Based on the $s_k^i$, a descriptive vector $d_i \in D$ in a given rally can be denoted as $(s_1^i, s_2^i, \ldots, s_n^i)$, where n is the length of strokes in this rally. For each rally, the sign of the end of the rally is that either side has lost a point, i.e., the loser has not returned the ball to the opponent's court, causing the stroke to not count.

Consider all the strokes, rallies, and games in each match, the later ones are more important than the earlier ones, as the later strokes, rallies, and games decide the result of the general results of each rally, game, and matches, respectively. Thus, if we consider that each match has a total score of 1, the scores for each game $g_j \in G$, each rally $r_i \in R_j$, and each stroke $s_k^i \in S_i$ can be calculated as follows, where $R_j$ denotes all the rallies in $g_j$, and $S_i$ denotes all the strokes in $r_i$.

$$Score(g_j) = \frac{2j}{|G| \cdot (|G| - 1)} \tag{1}$$

$$Score(r_i) = Score(g_j) \cdot \frac{2i}{|R_j| \cdot (|R_j| - 1)} \tag{2}$$

$$Score(s_k^i) = Score(r_i) \cdot \frac{2k}{|S_i| \cdot (|S_i| - 1)} \tag{3}$$

### 2.4. Evaluation Model of Tactical Benefit
#### 2.4.1. Tactical Benefit

According to the feature of a badminton match, i.e., each rally has only one result and each stroke has a limited start placement and destination, the players have limited choices to conduct their techniques and tactics. Thus, if we consider each single rally as an individual game with different weights, we can count all the rallies together, and compute the benefit of each stroke. Given a number of rallies $R = \{r_1, r_2, \ldots, r_m\}$, we construct a gaming tree $T$ for all the strokes $S_i = \{s_1^i, s_2^i, \ldots, s_n^i\}$ in each rally $r_i$. and compute the benefit for each node in the gaming tree. Specifically, each node of $T$ represent a possible stroke, and all the nodes of $T$ covers all the strokes in the selected rallies.

Figure 1 presents a simplified example (the specific technique and the destination for each stroke is not considered here) of building a gaming tree for three rallies $r_1, r_2, r_3$, where $S$ denotes strokes, $P$ denotes the player, the $Y$ denotes the placement, and $N$ denotes the gaming tree node. To illustrate, consider the strokes $\{s_1^1, s_1^2, s_1^3\}$, although they belong to different rallies, they are the first stroke for each rally and share the same player and placement (player $p_1$ with the placement 1), leading to the same tree node position $N_1$. Note that two strokes can be classified into the same tree node if and only if their player and placement are the same and their previous strokes (if existing) all have the same player and placement. For this reason, $s_2^1$ and $s_2^3$ can be classified to be node $N_2$, as they succeed $s_1^1$ and $s_1^3$ separately (both can be regarded as $N_1$) and have the same player and placement. Meanwhile, although $s_3^1$ and $s_3^2$ also have the same player and placement, they belong to different tree nodes as their previous strokes are different, i.e., $s_2^1$ and $s_2^2$ have different placement.

$S$: the stroke      $P$: the player
$Y$: the start position      $N$: the tree node

| Rally $r_1$ | | | | | Rally $r_2$ | | | | | Rally $r_3$ | | | |
|---|---|---|---|---|---|---|---|---|---|---|---|---|---|
| $S$ | $P$ | $Y$ | $N$ | | $S$ | $P$ | $Y$ | $N$ | | $S$ | $P$ | $Y$ | $N$ |
| $s_1^1$ | $p_1$ | 1 | $N_1$ | | $s_1^2$ | $p_1$ | 1 | $N_1$ | | $s_1^3$ | $p_1$ | 1 | $N_1$ |
| $s_2^1$ | $p_2$ | 3 | $N_2$ | | $s_2^2$ | $p_2$ | 9 | $N_3$ | | $s_2^3$ | $p_2$ | 3 | $N_2$ |
| $s_3^1$ | $p_1$ | 2 | $N_4$ | | $s_3^2$ | $p_1$ | 2 | $N_6$ | | $s_3^3$ | $p_1$ | 4 | $N_5$ |

**Gaming Tree** $\rightarrow$

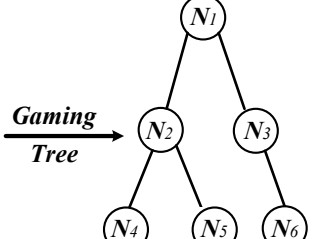

**Figure 1.** Illustration of the gaming tree for rallies.

Based on the gaming tree constructed from each stroke $s_k^i = (p_k^i, x_k^i, y_k^i, y_{k+1}^i)$, each tree node represents a possible stroke, and we can derive all the possible strokes for each single rally, as each rally starts from the root nodes (which have no predecessors) of the gaming tree and ends at leaf nodes (which have no successors). Meanwhile, the benefit of each possible stroke can be obtained by considering Formula (3) and the gaming tree, i.e., we summarize the score of each stroke $s_i$ that can be reduced to the same gaming tree node $N$ (denoted as $s_i \in N$) and regard it as the benefit of that stroke. Generally, the benefit of each Node N can be computed as follows.

$$Score(N) = \sum_{s_i \in N} Score(s_i) \cdot \begin{cases} 1 & \text{the server player wins} \\ -1 & \text{the server player loses} \end{cases} \tag{4}$$

For example, given two strokes $s_k^i$ and $s_k^j$ that can be reduced to the same gaming tree node $N$, and $s_k^i$ leads to the server player wins while $s_k^j$ is the opposite, we calculate the benefit of $N$ as $Score(N) = Score(s_k^i) - Score(s_k^j)$. Note that according to the Formulas (1)–(3), we can obtain that $\sum Score(s_k^i) = 1$. Thus, the benefit range of each node is $[-1, 1]$.

### 2.4.2. Evaluation Model

The evaluation model for badminton matches includes two steps, i.e., (i) constructing the gaming tree for the history strokes and evaluating the techniques and tactics for two players, and (ii) analyzing the rallies and strokes using the constructed gaming tree.

Evaluating. Given a set of existing matches, we first formulize each rally as $r_i = \{s_1^i, s_2^i, \ldots, s_n^i\}$ and each stroke as $s_k^i = (p_k^i, x_k^i, y_k^i, y_{k+1}^i)$. Then, we add $s_k^i$ into the gaming tree. For example, considering the rally $r_i = \{s_1^i, s_2^i, \ldots, s_n^i\}$, we first find if there exist a node $N$ that has the same $(p_1^i, x_1^i, y_1^i, y_2^i)$ as $s_1^i$. If not, we create a new node $N$ for $s_1^i$ in the gaming tree. After that, we update the $Score(N)$ with $Score(s_1^i)$. Next, we check the leaves of $N$ and check the following strokes $s_2^i, s_3^i, \ldots, s_n^i$ as $s_1^i$. After all the rallies and strokes are evaluated and the gaming tree is constructed, we compute the net benefit for each tree node as follows.

$$Net(N) = Score(N) + Max_{N_i \text{ is a leaf of } N}(Score(N_i)), \tag{5}$$

where $Max_{i \in m}(Score(N_i))$ denote the function that finds the maximal value among $\{Score(N_1), Score(N_2), \ldots, Score(N_m)\}$.

Consider the example shown in Figure 2, where four rallies are given, and the gaming tree is constructed with nine nodes. Specifically, in the first rally $r_1$, the server $p_1$ wins and continues to serve. In the second rally $r_2$, $p_1$ loses and alternates the service. However, the opponent player $p_2$ loses the rally $r_3$, and the service is alternated again. Finally, the player $p_1$ wins the fourth rally $r_4$ and the match ends. To model this match, we first build the gaming tree and compute the scores for each stroke, then the score of each tree node is obtained. Based on the above, we then compute the net benefit. Clearly, when it starts from $N_2$, the serve player always loses as the net benefit is negative.

*N*: the tree node   *Score*: importance of the stroke on the whole match

| Rally $r_1$ | | | | Rally $r_2$ | | | | Rally $r_3$ | | | | Rally $r_4$ | |
|---|---|---|---|---|---|---|---|---|---|---|---|---|---|
| *N* | *Score* | | | *N* | *Score* | | | *N* | *Score* | | | *N* | *Score* |
| $N_1$ | 0.0167 | | | $N_1$ | −0.02 | | | $N_2$ | −0.1 | | | $N_1$ | 0.0667 |
| $N_3$ | 0.0333 | | | $N_3$ | −0.04 | | | $N_4$ | −0.2 | | | $N_4$ | 0.133 |
| $N_6$ | 0.05 | | | $N_7$ | −0.06 | | | | | | | $N_8$ | 0.2 |
| | | | | $N_9$ | −0.08 | | | | | | | | |

**Figure 2.** Illustration of the gaming tree evaluation.

However, when it comes to starting from $N_1$, the tactics become complicated, as the opponent has two choices to play, i.e., move to $N_3$ with a benefit of −0.0067 or move to $N_4$ with a benefit of 0.133. According to the Nash Equilibrium, the opponent should move to $N_3$ to win this rally. Thus, although in this example the player that serves at $r_1$, $r_2$, and $r_4$ seems to have a higher probability of winning the match, the opponent also has the chance to change the result. This makes our proposed gaming tree not only capable of evaluating how the players have performed in the existing matches by leveraging the net benefit, but also capable of analyzing how the players use their techniques with tactics by finding Nash Equilibriums in the gaming tree.

Analysis. Based on the above, we find that the score of each stroke illustrates how important that stroke is to the whole match. Thus, we propose the following rating strategy to find Nash Equilibriums in the gaming tree, which helps analyze the existing strokes, rallies, games, and matches.

$$Score'(N) = \sum_{s_i \in N} Score(s_i) \tag{6}$$

Based on Equation (6), we compute the importance weight $Score'(N)$ of each node $N$. Next, we propose the win–loss flag table and the $Flg(N)$ function (depicted in Table 1) to determine whether the node $N$ leads to the server winning or losing. Note that the win–loss table is obtained by following the strategy that the player will choose the best strategy to win the game, and the opponent will make the player lose the game. To illustrate, consider a node $N$ that is an odd stroke, there are two situations: (i) $N$ is a leaf, the server must win when $Score(N)$ is positive, while the server loses if $Score(N)$ is negative, and the result is a draw if $Score(N) = 0$, and (ii) $N$ is non-leaf node, the server wins when it holds that $Max_{N_i \in successors\ of\ N}(Score'(N_i) \cdot Flg(N_i))$ is positive (the $Flg(N)$ function is defined in Table 1), and the result is a loss or draw if the value is negative or equal, respectively. Based on the above, we can also analyze the result when $N$ is an even stroke. The summarized table is shown below, which evaluates the result of each stroke (including odd and even strokes, leaf and no-leaf strokes) based on the importance functions $Score(\cdot)$ and $Score'(\cdot)$. For instance, if an odd stroke $N$ is a non-leaf node and it satisfies the condition that $Max_{N_i \in successors\ of\ N}(Score'(N_i) \cdot Flg(N_i)) > 0$, the stroke $N$ contributes to winning the match and the $Flg(N)$ is 1.

**Table 1.** Win–loss flag table.

| Node N | | Win/ Lose | Flg(N) |
|---|---|---|---|
| **Odd/Even** | **Situation** | | |
| Odd Stroke | Leaf node, and the $Score(N)$ is positive | Win | 1 |
| | Leaf node, and the $Score(N)$ is negative | Lose | −1 |
| | Leaf node, and the $Score(N)$ is 0 | - | 0 |
| | Non-leaf, $Max_{N_i \in successors\ of\ N}\left(Score'(N_i)\cdot\text{Flg}(N_i)\right) > 0$ | Win | 1 |
| | Non-leaf, $Max_{N_i \in successors\ of\ N}\left(Score'(N_i)\cdot\text{Flg}(N_i)\right) < 0$ | Lose | −1 |
| | Non-leaf, $Max_{N_i \in successors\ of\ N}\left(Score'(N_i)\cdot\text{Flg}(N_i)\right) = 0$ | - | 0 |
| Even Stroke | Leaf node, and the $Score(N)$ is positive | Win | 1 |
| | Leaf node, and the $Score(N)$ is negative | Lose | −1 |
| | Leaf node, and the $Score(N)$ is 0 | - | 0 |
| | Non-leaf, $Min_{N_i \in successors\ of\ N}\left(Score'(N_i)\cdot\text{Flg}(N_i)\right) > 0$ | Win | 1 |
| | Non-leaf, $Min_{N_i \in successors\ of\ N}\left(Score'(N_i)\cdot\text{Flg}(N_i)\right) < 0$ | Lose | −1 |
| | Non-leaf, $Min_{N_i \in successors\ of\ N}\left(Score'(N_i)\cdot\text{Flg}(N_i)\right) = 0$ | - | 0 |

Generally, based on the Win–loss flag table, the Nash Equilibrium for each tree node $N$ can be obtained, and the $Flg(N)$ corresponds to rally result (win, lose, or draw). Especially, we can predict whether the server will win or lose by directly checking the $Flg(\cdot)$ function for the root node (i.e., the initial stroke).

## 3. Results

### 3.1. Basic Data

The detailed information of the selected matches is shown in Table 2, with Lin winning 19 times and Lee winning 10 times in total. In Table 2, during the early period (2006 to 2009), the two players have roughly the same amount of wins. However, during the middle period (10–15), Lin showed a higher-level performance, and beat Lee in most matches. In the last three years (2016–2018), the two players came back to a stalemate.

**Table 2.** Match statics.

| No. | Year | Tournament | Match | Round | Winner |
|---|---|---|---|---|---|
| 1 | 2006 | Hong Kong Open | Super Series | Final | Lin |
| 2 | 2007 | Sudirman Cup | BWF tournaments | Group stage | Lee |
| 3 | 2007 | China Masters | Super Series | Semi-finals | Lin |
| 4 | 2007 | Japan Open | Super Series | Semi-finals | Lee |
| 5 | 2007 | Hong Kong Open | Super Series | Final | Lin |
| 6 | 2008 | Swiss Open | Super Series | Final | Lin |
| 7 | 2008 | Thomas Cup | BWF tournaments | Semi-finals | Lee |
| 8 | 2008 | Olympic Games | Multi-sport events | Final | Lin |
| 9 | 2008 | China Open | Super Series | Final | Lin |
| 10 | 2009 | All England Open | Super Series | Final | Lin |
| 11 | 2009 | Swiss Open | Super Series | Final | Lee |
| 12 | 2009 | Sudirman Cup | BWF tournaments | Semi-finals | Lin |
| 13 | 2010 | Thomas Cup | BWF tournaments | Semi-finals | Lin |
| 14 | 2010 | Japan Open | Super Series | Final | Lee |
| 15 | 2010 | Asian Games | Multi-sport events | Final | Lin |
| 16 | 2011 | All England Open | Super Series Premier | Final | Lee |
| 17 | 2011 | BWF World Championships | BWF tournaments | Final | Lin |
| 18 | 2011 | China Open | Super Series Premier | Semi-finals | Lin |
| 19 | 2012 | Korea Open | Super Series Premier | Final | Lee |
| 20 | 2012 | Olympic Games | Multi-sport events | Final | Lin |
| 21 | 2013 | BWF World Championships | BWF tournaments | Final | Lin |
| 22 | 2014 | Asian Games | Multi-sport events | Semi-finals | Lin |
| 23 | 2015 | Japan Open | Super Series | Last 16 | Lin |
| 24 | 2015 | China Open | Super Series Premier | Semi-finals | Lee |
| 25 | 2016 | Badminton Asia Championships | BAC tournaments | Semi-finals | Lee |
| 26 | 2016 | Olympic Games | Multi-sport events | Semi-finals | Lee |
| 27 | 2017 | Malaysia Open | Super Series Premier | Final | Lin |
| 28 | 2017 | Badminton Asia Championships | BAC tournaments | Semi-finals | Lin |
| 29 | 2018 | All England Open | Super 1000 | Quarter-finals | Lin |

### 3.2. General Analysis

In this paper, we analyzed the matches based off the gaming tree, while leveraging the benefit and net benefit (Formulas (4) and (5)) to evaluate the strokes of each player and used the Nash Equilibrium derived from the gaming tree and win–loss flags to analyze the general win–loss situation.

Firstly, according to the definition of the benefit, the higher benefit value implies that the stroke leads to higher contribution to winning the game. Thus, we computed the gaming tree for all the matches, and found the strokes with the top three benefits during the first three beats as shown in Figure 3, aiming at illustrating the tactics of two players for the first three beats. From the figures, we found that the best choice for Lin was to hit the shuttle to the backcourt while that for Lee was to control the forecourt.

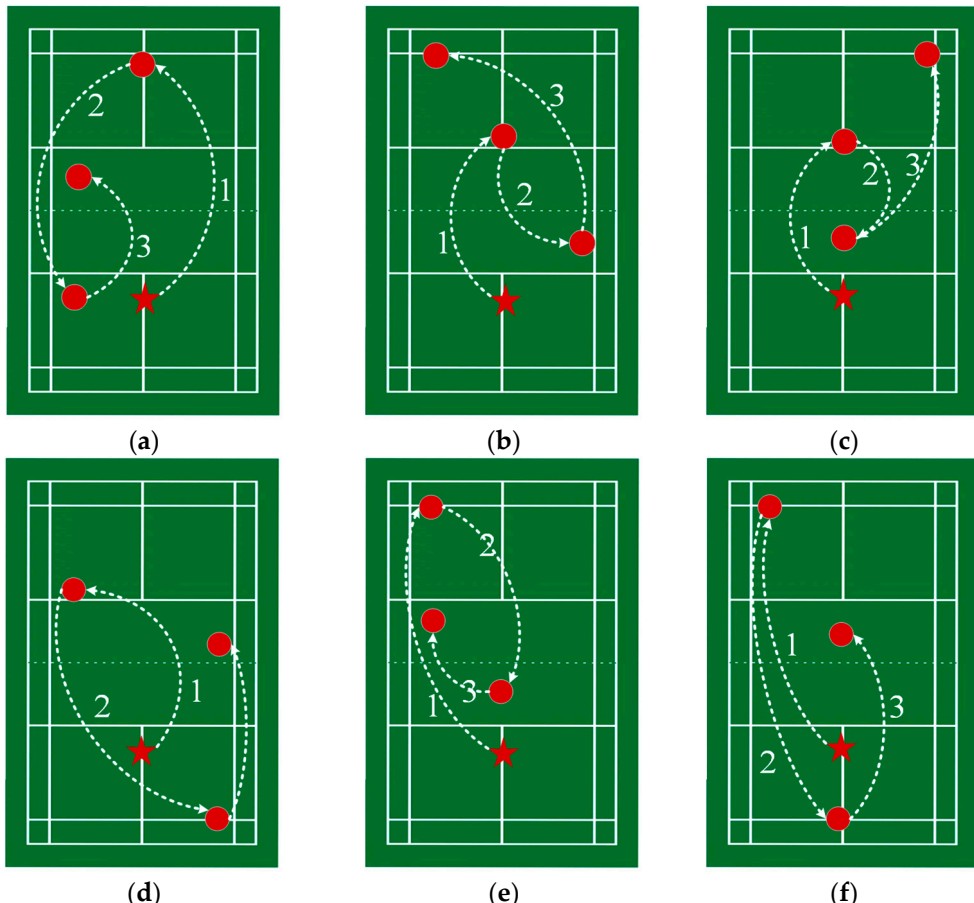

**Figure 3.** The top-three benefits strokes. (**a**) Lin's first stroke. (**b**) Lin's second stroke. (**c**) Lin's third stroke. (**d**) Lee's first stroke. (**e**) Lee's second stroke. (**f**) Lee's third stroke (the star points mean the start position of each stroke, i.e., the first stroke; the circle points denote the following positions, i.e., the second to the fourth stroke).

Next, we found the Nash Equilibrium based on the gaming tree, and derived the results as shown in Table 3. As can be seen from the table, with the growing beats, Lin and Lee have more choices to win or lose the match, while mostly the number of strokes that result in losing is larger than the strokes that lead to wining the rally. This also implies that Lin and Lee have many tactics and techniques to beat each other and will lose the game with high likelihood. However, during the matches, they defend with great skills and seize any opportunity to win the match. Additionally, according to the win–loss ratios, Lin has higher value than Lee, which is consistence with the results that Lin wins more matches than Lee.

**Table 3.** Win–Loss comparison for all matches.

|  | First 3 Beats | | First 5 Beats | | First 7 Beats | | First 9 Beats | | All Beats | |
|---|---|---|---|---|---|---|---|---|---|---|
|  | Win | Lose | Win | Lose | Win | Lose | Win | Lose | Win | Lose |
| Lin | 122 | 114 | 487 | 550 | 830 | 984 | 906 | 1066 | 910 | 1071 |
| Lee | 102 | 119 | 433 | 539 | 769 | 925 | 841 | 999 | 846 | 1005 |
|  | Win–Loss Ratio | | | | | | | | | |
| Lin | 0.517 | | 0.47 | | 0.458 | | 0.459 | | 0.459 | |
| Lee | 0.462 | | 0.445 | | 0.454 | | 0.457 | | 0.457 | |

### 3.3. Analysis for Different Periods

In different career periods, the results are different. Thus, we computed the Nash Equilibrium for different periods and presented the result in Figure 4. Specifically, we divided all the matches into four groups, i.e., matches 1–7, 8–14, 15–21, and 22–28. For each group, we constructed an individual gaming tree and computed their Nash Equilibrium results. The results are consistent with the analysis in Section 3.2, while there also exists a new observation that as the two players play more matches, they find more ways to beat each other. However, when it comes to the last period (matches 22–28), there was a significant drop for both players. This was because they were so familiar with each other that their tactics were no longer efficient, and their bodies could not support all their tactics and techniques. As a result, they tried to use the most effective way to win the game, making the number of gaming strategies decrease.

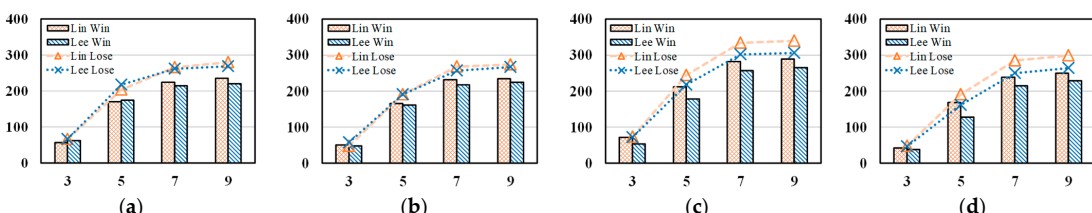

**Figure 4.** The Win–Loss results for different career period. (**a**) Matches 1–7. (**b**) Matches 8–14. (**c**) Matches 15–21. (**d**) Matches 22–28.

Additionally, we computed the win–loss ratio for two players in four career periods based on the win–loss results presented in Figure 4, and the results are shown in Figure 5. We divided all the 28 matches into four periods (1–7, 8–14, 15–21, 22–28) and let #x denote the x-th career period. For instance, #2 is the abbreviation for the second career period during the matches 8–14, while Lin#2 means the result for the player Lin during matches 8–14. In Figure 5, Lin#1 with parameter three denotes the win–loss ratio between the winning strokes and losing strokes for the player Lin among the first three beats in the career period #1 (i.e., the matches 1–7). We found that the results were consistence with the real matches, i.e., in the first three period the win–loss ratio of Lin was larger than Lee, thus Lin had more possibility of winning the game. In the last period, the ratio of Lin was larger than Lee's when the beats were not larger than five, while it becomes the opposite when the beats were seven or more. Thus, Lin and Lee had the equal possibility of winning the match.

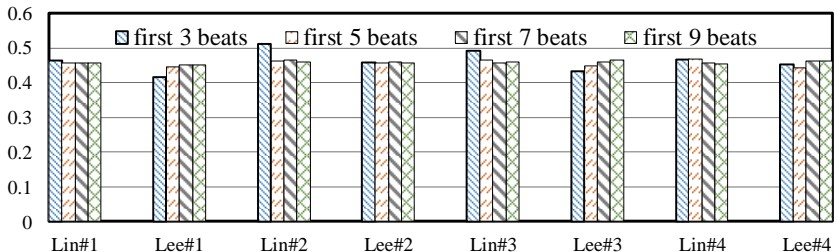

**Figure 5.** The Win–Loss ratio for different career period.

*3.4. Prediction Using Top-k Benefits*

This section may be divided by subheadings. It should provide a concise and precise description of the experimental results, their interpretation, as well as the experimental conclusions that can be drawn.

As stated in Section 2.4.1, the score and benefits imply how each stroke contributes to the match result. Thus, when the player comes to a specific node in the gaming tree, we can predict the next stroke by considering the top-k strokes with the highest benefits in the predecessors (children) of that node. For this reason, we constructed the gaming tree for four different career periods and used the strokes with top-k to predict the first m beats in the next match. Note that, we used four parameter groups of (k, m) to demonstrate the effectiveness of gaming tree prediction capability, i.e., P#1 (3, 3), P#2 (5, 3), P#3 (5, 5), and P#4 (5, 10). For example, for predicting the first period career of (matches 1–7) with parameters P#1 (3, 3), we used matches 1–6 to construct the gaming tree and used the strokes with top three benefits to predict all the first three beats.

The results are shown in Figure 6, where the Exist. (the abbreviation for existence) denotes the ratio of the existing strokes found in the gaming tree to the real number of strokes, and the Prec. (the abbreviation for precision) denotes the precision of correctly predicted strokes that exist in the gaming tree. As can be observed, the Exist. drops as the beats increase, while the precision generally keeps ascending, and the precision is above 90% when we use top-five benefit strokes to predict more than five beats, demonstrating the effectiveness of our strategy. Meanwhile, there is also an interesting observation that during the first period and the last period, when Lin and Lee have similar win–loss result, our model performs well for the first three beats on Lee. However, during the middle period (matches 8–21), when Lin wins more, the precision of our method for the first three beats on Lee is low. Thus, we can conclude that the first three beats of Lin are always hard to predict, while Lee can be predicted with high accuracy at the early and late career stage.

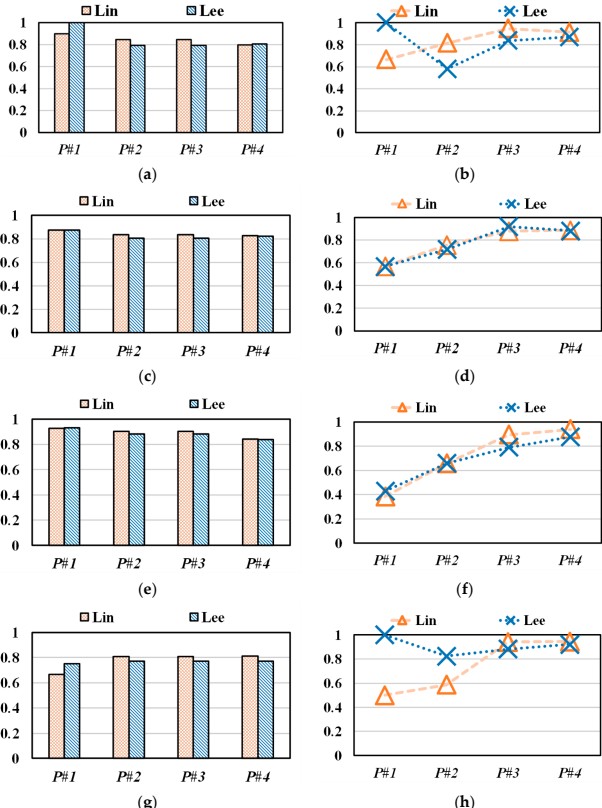

**Figure 6.** The prediction results for different career periods. (**a**) Exist. for matches 1–7. (**b**) Prec. for matches 1–7. (**c**) Exist. for matches 8–14. (**d**) Prec. for matches 8–14. (**e**) Exist. for matches 15–21. (**f**) Prec. for matches 15–21. (**g**) Exist. for matches 22–28. (**h**) Prec. for matches 22–28.

## 4. Discussion

Authors should discuss the results and how they can be interpreted from the perspective of previous studies and of the working hypotheses. The findings and their implications should be discussed in the broadest context possible. Future research directions may also be highlighted. In this paper, we used a gaming tree to analyze the technical and tactical strategies of Lin and Lee. Based on the analysis results, we made precise predictions for some of their matches. In addition, we analyzed the different stages of their professional careers and found that they have made some adjustments to their technical and tactical strategies with their increasing ages, thus revealing some hidden factors that determine the outcome of the match. However, with the recent adjustment of competition rules and the wide application of Hawk-Eye system, badminton matches are undergoing tremendous changes. The technical and tactical analysis revealed in this paper was only for top men's singles matches of the past decade.

There are still two aspects of technical and tactical analysis to be explored, i.e., Tactics combinations and Factors that are outside the tactics. Unlike male players, female players do not have strong offensive ability, resulting in a difference in the technical and tactical skills of female singles matches. Additionally, unlike single player matches, double matches require a higher cooperation ability from the players, resulting in the technical and tactical skills in doubles matches being completely different from those in singles matches. For instance, Cao et al. [30] identified special tactics in mixed double table tennis matches, while Abián-Vicén et al. [31] analyzed the different performance between men's and women's double matches.

Moreover, tactic length and tactic frequency also play key roles for players in controlling a match, as is illustrated by Liu [32] and Zhou [33]. This is because, when a match reaches its final stage, both players face increased pressure. Under such situations, using an efficient tactical combination (i.e., utilizing a smaller number of tactics) can significantly increase the chances of winning the match. Thus, tactical combinations are worth studying. In addition to the influence of tactical combinations, numerous factors that are outside tactics contribute to the outcome of badminton matches, such as the match length, the point difference, the stroke speed, and the Hawk-Eye system. Firstly, based on the analysis of recent math lengths, Iizuka et al. [34] suggested that badminton players strengthen their physical capabilities to win the match; Barreira et al. [35] found that a small point difference does not necessarily imply the winning of the match, while a difference of more than four points leads to a great possibility for winning the match; O'donoghue [36] observed the grand slam singles tennis matches and concluded that some key points determine the match results. As a result, such factors influence the direction of the matches and lead to different results.

In addition, spatial information on tactics and techniques also helps players win the match, as is illustrated by Chu et al. [37]. They demonstrated the significant influence of spatial information on tactics and techniques by visualizing badminton strokes, thus helping badminton players to win the match.

Meanwhile, speed, including stroke speed and movement speed, is also a contributing factor to the match result. For instance, a high-speed smash can quickly end a rally and earn points. However, in recent years, the advancement of racket manufacturing techniques and training programs have narrowed the gap in smash speed among top players. For example, in the Total Energies BWF Sudirman Cup Finals 2023, the smash speed of men's single players was around 400 km/h. Moreover, the BWF (Badminton World Federation) is dedicated to enhancing the viewing experience of the audience, and players can no longer win matches solely by fast smashes. Instead, movement speed has garnered more attention in recent years. For instance, Madsen et al. [38] developed a badminton-specific speed test for both skilled and normal players, and found that skilled players have significantly faster movement speed than normal players; Zhou et al. [39] studied the scoring rate of badminton technical movements in international competitions. However, precisely analyzing the relationship between match results and movement speed in real matches

remains a challenge and an open problem, due to the challenge of precisely obtaining movement speed data.

Moreover, with the recent use of the Hawk-Eye system in badminton matches, players are facing more challenges and matches become more attractive. For instance, once the Hawk-Eye system is called by players, the match pauses and the players can take a break and have the chance to reconsider the following tactic strategies thoroughly. Although there have been no studies on the influence of the Hawk-Eye system on badminton matches, recent studies have shown the great effect of Hawk-Eye system on the direction of tennis [40] and cricket [41] matches. However, due to the high cost of using the Hawk-Eye system, badminton players are only allowed to use the system at most two times in a match. Thus, the Hawk-Eye system has a limited influence on this study.

## 5. Conclusions

In this paper, we studied the problem of tactical benefit in badminton matches. To tackle this problem, we proposed a gaming tree to model the contribution of each stroke on the result of the match, and developed an evaluation model to obtain the Nash Equilibrium based on the existing matches. Specifically, given a number of existing strokes or rallies, our proposed method has the capability to obtain the most beneficial tactics based on the existing strokes, predict the most possible tactic combinations of the opponent player (e.g., predict the first several strokes of each rally or predict whether the player will win or lose the point at each move of the rally by computing the Nash Equilibrium for the current stage) and find the best tactics for scoring. Thus, our method could help coaches and players assess the benefits of their tactics in the existing matches, analyze the tactic strategies of opponent players while predicting their future strokes, and has a great potential in improving the players' strategies during the match by providing the most beneficial strokes based on existing rallies, which may change the match direction and contribute to improving the audience's experience.

**Author Contributions:** Conceptualization, W.L., Y.Z., W.G., X.W. and S.Y.; methodology, Y.Z. and S.Y.; software, W.L., W.G. and X.W.; validation, W.G. and X.W.; formal analysis, W.L. and Y.Z.; investigation, W.G. and X.W.; resources, S.Y.; data curation, Y.Z.; writing—original draft preparation, W.L. and Y.Z.; writing—review and editing, W.G., X.W. and S.Y.; visualization, Y.Z.; supervision, W.L.; project administration, W.G., X.W. and S.Y. All authors have read and agreed to the published version of the manuscript.

**Funding:** This research received no external funding.

**Institutional Review Board Statement:** Not applicable.

**Informed Consent Statement:** Not applicable.

**Data Availability Statement:** Publicly available videos of badminton matches in official competitions were used to analyze the data. For this purpose, we used common video platforms (e.g., youtube.com) (accessed on 25 January 2022).

**Conflicts of Interest:** The authors declare no conflict of interest.

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
