# Peer review of "Gaming Tree Based Evaluation Model for Badminton Tactic Benefit Analysis and Prediction"

_applsci, doi:10.3390/app13137380_

Round 1

Reviewer 1 Report

The paper presents a very interesting and novel application of gaming trees and the Nash equilibrium. The paper is well written and the authors explained most of the concepts well. I would like to see a bit more, high level, explanation of the Nash equilibrium and context thereof within this application. I believe there is a lot of value in the presented methodology for other sports as well.

Some corrections required:

Line 68; a period '.' is required after Internet.

Line 94; insert 'the' before rally.

Line 99; full stop after rallies, and begin new sentence with: A stroke ...

Line 100; The x_{k+1}^i should be y_{k+1}^i. Fix this throughout the paper.

Line 114; ... of a badminton match ...

Line 126; rallies

Line 129; placement are the same ...

Line 132 - 134; the statement is wrong: the strokes referred to has different placements. Revise and correct the statement.

Line 147; the superscript should be j in the second score function.

Line 176; 0.133

Line 177; serves

Line 193; loses

Line 193; is a draw ... 

Line 194; the function FLG is not defined.

Line 208; at most matches.

Line 209; years 

Line 220; beats

Line 229; defend with great 

Line 236; periods

Line 239; matches

Line 265; periods

Line 272; denotes the number of

Figure 6; the number of strokes is not shown on the figure as a second y-axis? I suggest splitting the figures to show only the precision of the predictions on one figure. T precision for the different career periods can be shown on one figure. The sub-figures have the same numbers. The figure captions are wrong. 

Line 321; propose a gaming tree

Line 321; considers

In the comments to authors

Reviewer 2 Report

Dear authors, thank you very much for the excellent work presented, but I have some questions and contributions to it.

1.       Pag - 58 to 60: "the objective of the study should be: ............"

2.       Page 63: delete "In this paper".

3.       STROKE SPEED: the author indicates that it was not contemplated yet states that it is an important variable, I consider that in the discussion it should be placed a section on limitations and this factor should be placed as a limitation explaining why.

4.       Figure 1: A legend should be put with what is in the figure, namely the letters S, P, Y and N as well as the remaining characters.

5.       Figure 2: A legend should be added.

6.       Talk about NASH EQUILIBRIUM, I consider that the concept should be introduced in the introduction since the reader has not had any contact with the concept until his first report here (after FIG 2).

7.       At the end of table 1, there is a comment that should be turned into a caption and the components of the table should be shown.

8.       Figure 5: Caption with Lim and Lee #1 to 4 and captions 3, 5, 7 and 9

9.       page 261 - put section or point (not 'SEC.')

10.   Figure 6 - Legend P#1 to 4, legend "exis" and "prec" and put name and units on vertical axis.

11.   Pages 272 and 273 - if you put EXIST and PREC, you must say what each is.

12.   EXIS and EXIST - a different term (I think) is used to describe the same thing, it MUST ALWAYS BE EQUAL.

13.   Discussion: the discussion is divided into only 2 points, I consider the discussion weak and believe that it can be greatly improved, and take into consideration all aspects covered in the article, the discussion should not be divided, it should be followed and ideas should be chained logically.

14.   The authors talk about HAWK-EYE SYSTEM: this system was implemented in 2014 and the data analysis is from 2006 to 2018.

·         How does it influence the game?

·         Did it influence the study?

·         WHY?

15.   Conclusions: from line 320 to 325 everything written is redundant (it has been said before) conclusions should be concrete answers to the results of the proposed objectives. The conclusion should be reworded

16. GENERAL QUESTION: Do you think your model can make the game predictable? That is, at each move, the player knows in advance how the point will end and who will win and/or lose?

Round 2

Reviewer 2 Report

Dear authors, thank you very much for your work.

I believe that the article is now much better and accessible to the reader.